# GARP Regulates the Immune Capacity of a Human Autologous Platelet Concentrate

**DOI:** 10.3390/biomedicines10123136

**Published:** 2022-12-05

**Authors:** Emily R. Trzeciak, Niklas Zimmer, Peer W. Kämmerer, Daniel Thiem, Bilal Al-Nawas, Andrea Tuettenberg, Sebastian Blatt

**Affiliations:** 1Department of Dermatology, University Medical Center Mainz, Johannes Gutenberg University Mainz, 55131 Mainz, Rhineland-Palatinate, Germany; 2Department of Oral and Maxillofacial Surgery, University Medical Center Mainz, Johannes Gutenberg University Mainz, 55131 Mainz, Rhineland-Palatinate, Germany; 3Research Center for Immunotherapy, University Medical Center Mainz, Johannes Gutenberg University Mainz, 55131 Mainz, Rhineland-Palatinate, Germany; 4Platform for Biomaterial Research, BiomaTiCS Group, University Medical Center Mainz, Johannes Gutenberg University Mainz, 55131 Mainz, Rhineland-Palatinate, Germany

**Keywords:** iPRF, liquid platelet rich fibrin, autologous platelet concentrate, wound healing, GARP, TGF-ß, platelets, T cells, regulatory T cells, macrophages

## Abstract

Autologous platelet concentrates, like liquid platelet rich fibrin (iPRF), optimize wound healing; however, the underlying immunological mechanisms are poorly understood. Platelets, the main cellular component of iPRF, highly express the protein, Glycoprotein A repetitions predominant (GARP), on their surfaces. GARP plays a crucial role in maintaining peripheral tolerance, but its influence on the immune capacity of iPRF remains unclear. This study analyzed the interaction of iPRF with immune cells implicated in the wound healing process (human monocyte derived macrophages and CD4^+^ T cells) and evaluated the distinct influence of GARP on these mechanisms in vitro. GARP was determined to be expressed on the surface of platelets and to exist as a soluble factor in iPRF. Platelets derived from iPRF and iPRF itself induced a regulatory phenotype in CD4^+^ T cells, shown by increased expression of Foxp3 and GARP as well as decreased production of IL-2 and IFN-γ. Application of an anti-GARP antibody reversed these effects. Additionally, iPRF polarized macrophages to a “M0/M2-like” phenotype in a GARP independent manner. Altogether, this study demonstrated for the first time that the immune capacity of iPRF is mediated in part by GARP and its ability to induce regulatory CD4^+^ T cells.

## 1. Introduction

Autologous platelet concentrates (APC) improve wound healing and tissue regeneration. APC are defined as the volume fraction of plasma in autologous whole blood that has a (up to five times) higher thrombocyte concentration than regular blood samples and contains platelet related growth factors, which promote wound healing and tissue repair [1]. APC are widely used in regenerative surgery. Clinical indications include but are not limited to orthopedic procedures as well as dentistry and oral surgery [2,3]. Important representatives of the first generation of APC are platelet rich plasma (PRP), preparation rich in growth factor (PRGF), or pure-PRP (P-PRP), that are still widely used in dental and medical reconstructive procedures [4,5]. They are produced chair side via a specific centrifugation process. After venous blood collection with respective vacutainers and tube systems, an anticoagulant is added before and/or after centrifugation. In this way, thrombocytes can be extracted within a fibrin clot and used for different medical applications [6,7,8,9].

The main disadvantages of this preparation are the usage of anticoagulants that counteract the autologous concept and the exclusion of leukocytes and other cells important for wound healing in the (often multi-step) centrifugation process, which may lead to unfavorable outcomes [10]. To overcome these limitations, second generation APC, such as platelet rich fibrin (PRF), were developed [11]. Here, only one centrifugation step is needed without the addition of an anticoagulant [12]. Depending on the protocol, either a stable clot or a liquid fibrin solution (iPRF) can be produced, that contains potentially higher thrombocyte concentrations and growth factor levels in vitro in comparison to first generation APC [13]. Entangled components of the resulting fibrin net of PRF include different cells, like thrombocytes and leukocytes, and growth factors, which are slowly released over time that can interact with other cell types, initiate angiogenesis, and are implicated in other processes involved in the wound healing phase [11,13,14].

Although APC are recognized for their wound healing properties and widely used in regenerative medicine, little is known about the immune capacity of APC and the possibly involved cellular mechanisms [15,16,17]. Initial studies have provided striking data that APC directly influences host immune responses via macrophage polarization [16,18]. Recent evidence has also arisen that platelets, the principle cellular component of PRF, play essential roles in innate and adaptive immunity via their interaction with different immune cells [19].

Glycoprotein A repetitions predominant (GARP) is a surface docking receptor and activator of latent transforming growth factor beta 1 (LTGF-β1) that is enriched on the surface of platelets [20]. Released primarily from platelets upon tissue injury, TGF-β1 plays an essential role in the wound healing process, mediating inflammation, angiogenesis, and connective tissue regeneration, amongst other things [21,22,23]. GARP has been shown to exhibit anti-inflammatory properties itself through both TGF-β1 independent and dependent mechanisms. In more detail, GARP can be cleaved from the surface of platelets as a soluble factor (sGARP), facilitating the release of active TGF-β1 into the surrounding environment. Platelet derived GARP was also observed to exhibit potent anti-inflammatory functions, including the induction of Foxp3^+^ regulatory T cells (Treg) as well as the inhibition of T effector cells [24,25]. Despite these desirable properties, which could be used to prevent overshooting inflammatory responses during wound healing, coupled with the high expression of GARP on platelets, a major component of PRF, GARP has not yet been examined on how it might influence the wound healing properties of APC.

This study aimed to analyze the impact of iPRF on the phenotype and function of cells implicated in the wound healing response, namely human monocyte derived macrophages and CD4^+^ T cells, and to determine the distinct influences of GARP on these mechanisms in vitro. CD4^+^ T cells and macrophages were evaluated as they are known to migrate to the site of injury, where iPRF is applied to in patients, and play essential roles in mediating the wound healing response [24,26,27,28]. To determine the direct influence of iPRF on CD4^+^ T cells and macrophages, the respective cell types were cocultivated with iPRF/iPRF derived platelets. This study utilized iPRF as a model for second generation APC as it is known to contain an enriched amount of growth factors that are slowly released overtime and leukocytes, which may optimize wound healing [11,13,14].

GARP was found to be expressed on the surface of platelets in iPRF and to exist as a soluble factor in iPRF. iPRF derived platelets and iPRF could induce a regulatory phenotype in CD4^+^ T cells through a GARP dependent mechanism. In more detail, heightened levels of Foxp3 and GARP as well as decreased effector cytokine production was observed in CD4^+^ T cells treated with iPRF derived platelets or iPRF. These effects could be reversed with a blocking anti-GARP antibody. It was also shown that iPRF induces a wound healing “M0/M2 like” phenotype in macrophages via a GARP independent mechanism. Altogether, these results identified for the first time that the immune capacity of iPRF is mediated in part by GARP and its ability to induce Treg. These findings contribute to our understanding of the underlying immunological mechanisms that occur during the wound healing process following iPRF treatment. They may also be applied to therapeutically enhance the wound healing properties of APC via the addition of GARP to new concentrates.

## 2. Materials and Methods

### 2.1. Preparation of iPRF

The study was conducted in accordance with the Declaration of Helsinki, and the protocol was approved by the Ethics Committee of the Landesärztekammer Rhineland-Palatine (no. 2019-14705_1, approved 9 September 2020). Informed consent was obtained from all participants involved in the study. In more detail, iPRF from healthy donors was manufactured as described in the literature [29,30]. Briefly, venous blood collection of 60 mL via special vacutainer systems (Process for PRF, i-PRF, Nice, France) was performed. Next, blood was immediately centrifuged according to the manufacturer’s protocol (700 rpm for 3 min, relative centrifugal force (rcf-max) = 60× *g* at 40° rotor angulation with a radius of 88 mm at the clot and 110 mm at the max; Process for PRF, Duo Quattro centrifuge). After centrifugation, the yellow upper part (approximately 2 mL of each tube) was collected as iPRF, and the underlying parts containing red blood cells were discarded. iPRF was used as subsequently described. Healthy donors used in this study (n = 3) consisted of two males and a female between the ages of 30–36. Donors were on no medications, had no allergies, no severe illnesses, or history of smoking/tobacco use.

### 2.2. Isolation of Platelets from iPRF

iPRF was obtained from healthy donors as described above. Subsequently, platelets were isolated from iPRF as previously described [24]. To remove any potential leukocyte contamination, iPRF was collected and centrifuged for an additional 15 min at 200× *g* at RT. The resulting platelet rich supernatant was then centrifuged for 5 min at 2000× *g* at RT, after being rinsed with 1× PBS at a ratio of 1:1. Platelets from the resulting pellet were counted and used as subsequently described.

### 2.3. Isolation and Stimulation of Human CD4^+^ T Cells

As described earlier, CD4^+^ CD25^−^ T cells were isolated from human buffy coats and stimulated with anti-CD3 and anti-CD28 antibodies [24]. Buffy coats were provided by healthy donors. This study was performed in agreement with the Declaration of Helsinki, and the Ethics Committee of the Landesärztekammer Rhineland-Palatine approved the protocol (no. 837.019.10 (7028), approved 4 March 2010). Isolated platelets or iPRF were added to CD4^+^ T cells cultures at a ratio of 50:1 (Platelets/iPRF:CD4^+^ T cell) as indicated at day 0 to form cocultures, meaning two different cell types were grown together as a way to examine their direct influence on each other. Cocultures were either left untreated or treated with 10 µg/mL anti-GARP antibody (Ab) (Origene, #AP17415PU-N, Rockland, MD, USA), 10 µg/mL anti-TGF-β receptor II Ab (R&D Systems, #AF-241-NA, Minneapolis, MN, USA), and 10 µg/mL anti-TGF-β I-III Ab (R&D Systems, #MAB1835R) at day 0 as indicated.

Carboxyfluorescein succinimidyl ester (CFSE) (eBioscience #65-0850-84, San Diego, CA, USA) was used to prelabel CD4^+^ T cells to track their proliferation. Labeled CD4^+^ T cells were plated in 48 well plates at 10^6^ cells/mL in X-Vivo 15 (Lonza, #BE02-060F, Basel, Switzerland) and incubated for 3 days in the presence or absence of platelets derived from iPRF at a ratio of 1:50 as well as 10 µg/mL anti-GARP Ab (Origene) before undergoing flow cytometric analysis.

### 2.4. Isolation and Polarization of Human Monocyte Derived Macrophages

Buffy coats were obtained from healthy volunteers, and macrophages were isolated from them as previously described [31]. In brief, 150 million peripheral blood mononuclear cells (PBMC) were plated in RPMI1640 (Thermo Fisher Scientific, #31870, Waltham, MA, USA), containing 1% Glutamax (Thermo Fisher Scientific, #35050038) and 0.1% Primocin (InvivoGen, #ant-pm-2, San Diego, CA, USA) in cell culture Petri dishes (Corning, #430167, Corning, NY, USA) for at least 1 h in a 37 °C 5% CO_2_ incubator to promote attachment. Non-adherent cells were washed away with repeated pre-warmed phosphate-buffered saline (PBS) washes (Thermo Fisher Scientific, #14190-094). Adherent cells were cultured in the medium described above containing an additional 1% heat inactivated human plasma and 50 ng/mL M-CSF (ImmunoTools, #11343113, Friesoythe, Germany). Generation of heat inactivated human plasma was prepared as previously described [31]. After 5–7 days, “M0” macrophages were collected with Accutase (Thermo Fisher Scientific, #00-4555-56) and replated at 10^6^ cells/well in 6 well plates (Corning, #3506). Cells underwent polarization to an “M1-like” or a “M2-like” phenotype as previously described [31]. For polarization experiments, “M0-like” macrophages were treated with 400 µL of iPRF/well. Cells were polarized for 2 days before undergoing flow cytometric analysis.

### 2.5. Flow Cytometry

Single cell suspensions were prepared and stained for flow cytometric analysis. Fibrin clots, resulting from iPRF, were manually disrupted by mashing through a 40 μm cell strainer (VWR International, #732-2757, Radnor, PA, USA) to generate single cell suspensions. This enabled the recovery of both CD4^+^ T cells and macrophages that had become entangled in the fibrin clot. Cell suspensions were washed with prechilled PBS before undergoing further analysis.

Isolated platelets were stained with anti-CD41a (eBioscience #11-0419-42) and anti-GARP (Miltenyi Biotec, #130-103-820, Bergisch Gladbach, Germany) antibodies as well as the respective IgG1 isotype control antibody (Miltenyi Biotec, #130-113-438).

For intranuclear staining of Foxp3, CD4^+^ T cells were harvested after 3 days of coincubation, and they were stained with fixable viability dye (Invitrogen, #65-0865-14, Waltham, MA, USA), followed by the surface staining with anti-GARP (Miltenyi Biotec, #130-103-820). Cells were fixed, and permeabilized with the Foxp3/Transcription Factor Staining Buffer Kit (eBioscience, #00-5523-00), followed by staining with anti-Foxp3 (BioLegend, #320208, San Diego, CA, USA). After 6 days of coincubation, CD4^+^ T cells were stimulated for 5 h with 50 ng/mL phorbol 12-myristate 13-acetate (PMA) (Sigma Aldrich, #P1585-1MG, Munich, Germany), 1 µg/mL ionomycin (Enzo Life Sciences, #ALX-450-006-M001, Lörrach, Germany), and 1.3 µM monensin (BD, #554724, Heidelberg, Germany). Cells were stained with fixable viability dye (Invitrogen, #65-0865-14), fixed, and permeabilized with BD Cytofix/Cytoperm Plus (BD Biosciences, #555028, Heidelberg, Germany), followed by intracellular staining with anti-IFN-γ (BD Biosciences, #557643) or anti-IL-2 antibodies (eBioscience, #17-7049-42).

Macrophages were stained with fixable viability dye (Invitrogen, #65-0866-14) followed by surface antibody staining of anti-CD40 (Immunotools, #21270403), anti-CD36 (Miltenyi Biotec, #130-095-480), anti-CD206 (Miltenyi Biotec, #130-100-152), anti-PD-L2 (Miltenyi Biotec, #130-105-829), anti-CD163 (Life Technologies, Carlsbad, CA, USA, #A15792), anti-HLA-DR (MHC II) (BioLegend, #307650), and anti-CD14 (Invitrogen, #48-0149-42). Measurements were performed on a BD LSRII flow cytometer (BD Biosciences) and were analyzed with Cytobank software [32]. Doublets, debris, and dead cells were excluded from analysis. Example gating strategies of the different cell types examined in this study can be found in Appendix A.

### 2.6. Western Blot

Protein lysates of iPRF were prepared by diluting iPRF twenty-fold with the following cell lysis buffer, consisting of 20 mM TRIS-HCl pH 7.5, 150 mM NaCl, 1% Triton X-100, 1 mM Na_2_-EDTA, 1 mM EGTA, 1 mM β-glycerophosphate, 2 M urea, and 1x protease/phosphatase inhibitor cocktail (Thermo Fisher Scientific, #78440). Samples were incubated in lysis buffer for 10 min, followed by brief sonication. Lysates were loaded into reducing SDS page gels at 10–20 μg/well as indicated. Human recombinant soluble GARP (R&D Systems, #6055-LR-050) was used as a positive control for antibody specificity and loaded at 1 ng/lane. Western blot was carried out using the NuPAGE system (Thermo Fisher Scientific), in accordance with the manufacturer’s recommendations. Membranes were probed with the following antibodies: anti-GARP (Cell Signaling, #83565, Danvers, MA, USA), anti-β-actin (Cell Signaling, #8457), and anti-rabbit horseradish peroxidase (Jackson Immunoresearch, #211-032-171, West Grove, PA, USA).

### 2.7. GARP ELISA

iPRF was analyzed for its soluble GARP content by enzyme-linked immunosorbent assay (ELISA), according to the manufacturer’s protocol (R&D Systems #DY6055). Samples were diluted 1:1 with Reagent Diluent.

### 2.8. Statistics

Data was analyzed by GraphPad Prism (GraphPad Software, version 8.0.0 for Windows, San Diego, CA, USA, www.graphpad.com, accessed on 1 October 2022). Results were normalized to the untreated (*w/o*) controls as indicated. Data show mean ± standard deviation (SD). Statistical significance was calculated by one-way ANOVA, two-way ANOVA, and unpaired Student’s *t*-tests as listed in the figure legends. Statistical significance is indicated by asterisks as follows: * *p* < 0.05, ** *p* ≤ 0.01, *** *p* ≤ 0.001, **** *p* ≤ 0.0001 and not significant (n.s.) *p* > 0.05.

## 3. Results

### 3.1. Glycoprotein A Repetitions Predominant (GARP) Is Expressed on the Surface of Platelets Isolated from iPRF and Exists as a Soluble Factor within iPRF

GARP was first characterized to be expressed on the surface of platelets, and activated platelets are known to upregulate GARP expression on their surfaces [20]. However, up until now, no study has shown that GARP is expressed in autologous platelet concentrates. Here, it was demonstrated for the first time that GARP is expressed on the surface of platelets found in freshly generated iPRF. By flow cytometric analysis, it was determined that approximately half the platelets expressed GARP on their surfaces, and the expression of GARP was highly elevated in comparison to the isotype control (Figure 1A,B). Previous work, using pre-activated platelets, showed comparable percentages of GARP^+^ platelets and GARP expression levels [24]. This suggests that part of the platelets found in iPRF could be partially activated, presumably resulting from their preparation process, but this should be validated in future studies.

These results could be validated by Western blot, confirming the presence of GARP in iPRF (Figure 1C). Interestingly, it was found that GARP also exists in its soluble form (sGARP) in both iPRF and peripheral blood, likely released from the surface of activated platelets (Figure 1D). However, sGARP levels were not found to be significantly increased in comparison to peripheral blood (Figure 1D). Altogether, these results indicate that GARP is expressed in iPRF, and it exists as both a surface protein on platelets as well as a soluble factor inside of iPRF.

### 3.2. iPRF Derived Platelets Induce a Regulatory Phenotype in CD4^+^ T Cells via a GARP Dependent Mechanism

Previously, it was shown that GARP derived from platelets exhibits potent tolerogenic properties, including the induction of peripheral Treg [24]. To determine if these effects are conserved in iPRF, platelets were isolated from iPRF and cocultured with CD4^+^ T cells. It was found that CD4^+^ T cells had a significantly increased expression of Foxp3 in the presence of platelets on day 3 of coincubation (*p* < 0.05) (Figure 2A). This effect could be reversed back to untreated CD4^+^ T cell levels with the addition of an anti-GARP antibody (no significant difference vs. *w/o*) (Figure 2A). Furthermore, on day 3 of coincubation, the addition of iPRF derived platelets increased the GARP expression of CD4^+^ T cells (*p* < 0.001); however, the proliferation of iPRF treated CD4^+^ T cells remained unaffected (no significant difference vs. *w/o*) (Figure 2B,C). Blocking with an anti-GARP antibody could not restore proliferation and GARP levels to the untreated CD4^+^ T cell control (no significant difference vs. *w/o*, *p* < 0.001) (Figure 2B,C).

On day 6 of co-incubation, production of the effector cytokines, IFN-γ and IL-2, by CD4^+^ T cells was analyzed. For both cytokines, production was significantly decreased, when CD4^+^ T cells were cocultured with iPRF derived platelets (*p* < 0.01 and *p* < 0.05) (Figure 2D,E). Once more, it was observed that the addition of an anti-GARP antibody could completely reverse these effects to untreated CD4^+^ T cell levels (no significant difference vs. *w/o*) (Figure 2D,E). Collectively, these findings show that platelets, isolated from iPRF, induce a regulatory phenotype in CD4^+^ T cells and are capable of suppressing T effector cell function.

### 3.3. iPRF Induces a Regulatory Phenotype into CD4^+^ T Cells

As mentioned earlier, iPRF contains a mixture of different cell types and their respective growth factors [11,13,14]. To better ascertain the collective influence of iPRF on CD4^+^ T cells, cocultures of iPRF and CD4^+^ T cells were performed. In agreement with the results above (Figure 2A), Foxp3 expression was found to be significantly increased on day 3 of coincubation (*p* < 0.01), and application of an anti-GARP antibody was able to reverse this effect (*p* < 0.01) to untreated CD4^+^ T cell levels (Figure 3A). Additionally, effector cytokine production of IL-2 and IFN-γ by CD4^+^ T cells on day 6 of coincubation revealed a significant decrease (each *p* < 0.0001), when CD4^+^ T cells were cocultured with iPRF (Figure 3B,C). Treatment with an anti-GARP antibody was able to partially restore IL-2 production (*p* < 0.01) and could completely restore IFN-γ (no significant difference vs. *w/o*) levels to the untreated CD4^+^ T cell control (Figure 3B,C).

The tolerogenic properties of GARP are known to be partially mediated through a TGF-ß dependent mechanism [20,24]. To clarify whether the observed regulatory effects of iPRF on CD4^+^ T cells occur in a TGF-ß independent or dependent manner, cocultures of CD4^+^ T cells and iPRF were treated with anti-TGF-ß receptor II (anti-TGF-ß-RII) and anti-TGF-ß-I-III blocking antibodies. Interestingly, in contrast to the effects of anti-GARP antibody treatment, anti-TGF-ß-RII and anti-TGF-ß-I-III antibodies did not show any noteworthy influence on Foxp3 expression (no significant difference vs. *w/o*) and were unable to restore effector cytokine production of IL-2 and IFN-γ to untreated levels (all *p* < 0.0001) (Figure 3A–C). In summary, iPRF induced a regulatory phenotype in CD4^+^ T cells via GARP in a seemingly TGF-ß independent manner.

### 3.4. iPRF Polarizes M0-Macrophages into a More “M0/M2-like” Phenotype in a GARP Independent Manner

Previously, it was shown that APC can polarize macrophages towards a wound healing “M2-like” phenotype [16,17,18,33]. To build on these results, by staining with an expanded surface marker panel, human monocyte derived M0 macrophages were treated with iPRF and analyzed for altered surface marker expression via flow cytometry. It was demonstrated that iPRF induces a “M0/M2-like” phenotype as shown by the significantly increased expression of the M2 surface marker, CD36 (*p* < 0.0001) and corresponding decreased expression of the M1 surface markers, CD14 (*p* < 0.0001) and CD163 (*p* < 0.01) (Figure 4), thus validating and expanding upon the results of previous studies [16,17,18,33].

sGARP has also been reported to induce M2 macrophage polarization, and it was confirmed to be expressed in iPRF (Figure 1D) [34]. Therefore, it was investigated if GARP may play a role in the underlying immunological mechanism behind the induction of “M0/M2-like” macrophages following iPRF treatment. To investigate this, macrophages were cocultured with iPRF and were treated with a blocking anti-GARP antibody. Interestingly, unlike CD4^+^ T cells, anti-GARP antibody treatment was unable to reverse the effects of iPRF on macrophage polarization. This indicates that a GARP independent mechanism appears to play a greater role in the influence of iPRF on macrophage polarization.

## 4. Discussion

This study is the first to show the direct interaction between the APC, iPRF, and immune cells via the immunosuppressive protein, GARP. As a major result, it was shown that application of iPRF and iPRF derived platelets induced a regulatory phenotype in CD4^+^ T cells through a GARP dependent mechanism (Figure 2 and Figure 3). Blocking GARP with an anti-GARP antibody could effectively reduce Foxp3 expression and effector cytokine production to untreated CD4^+^ T cell levels, providing a potential targeting method. These findings are in accordance with the literature, where it was previously demonstrated that platelet derived GARP could induce peripheral Treg [24]. In addition, it was observed that the regulatory effects of iPRF and platelets derived from iPRF on CD4^+^ T cells were nearly identical—indicating that platelets are a key cell type in iPRF, which influences its immunological capacity. Lastly, platelet lysates are known to have an impact on various immune cell types, including T cells and macrophages, thus further supporting their important role in the immune capacity of APC [35].

Treg have become increasingly recognized to orchestrate tissue repair and regeneration. Recruited to the damaged site, Treg can indirectly modulate regeneration by controlling neutrophils and helper T cells, inducing M2 macrophage polarization, and directly facilitating regeneration via the activation of progenitor cells at the site of tissue damage to mediate inflammation resolution and to regulate immunity after injury [28]. Weirather et al., 2014 could show that depletion of Treg in a post-myocardial infarction model led to an inhibited production of anti-inflammatory factors essential for wound healing, like IL-10 and TGF-β1, an increased infiltration of neutrophils and monocytes, and an induction of “M1-like” macrophages [36]. Whereas therapeutic activation of Treg via an anti-CD28 monoclonal antibody led to an accumulation of “M0/M2-like” macrophages and exhibited enhanced wound healing. The discovery that iPRF induces CD4^+^ Treg via GARP and that these Treg display heightened levels of GARP, a known marker of activated Treg, as well as increased suppressive function, help to explain in part a mechanism behind the wound healing properties of iPRF.

So far, only limited evidence exists regarding the immune capacity of APC, especially for PRF. An anti-inflammatory activity has been proven, yet the exact mechanism remained unclear [17]. In previous studies, it was shown that human PRF can interact with macrophages [16,17,18,33]. Here, PRF not only suppressed the M1 pro-inflammatory status of the macrophages but also supported their shift towards the M2 resolving phenotype [17,18]. These findings are encouraging as M2 macrophages have been shown to be beneficial in transplant acceptance, wound healing, and tissue remodeling, helping to explain part of the observed wound healing properties of iPRF [37]. So far, only one other known study has shown that PRP can polarize human monocyte derived macrophages to a “M2-like” phenotype [18]. The findings of this study are in accordance with our results. However, it is important to note that this work characterized the effects of iPRF, a differentially prepared APC, on the polarization of human monocyte derived macrophage for the first time and provided a more detailed analysis of macrophage surface marker alterations following iPRF treatment. Furthermore, this work also evaluated GARP as a possible underlying mechanism for the observed effects of iPRF on macrophage polarization.

Previously, it was shown by Hahn et al., 2016 that application of sGARP could polarize macrophages to a “M0/M2-like” wound healing phenotype [34]. Contrary to these findings, application of a blocking anti-GARP antibody was unable to effectively reverse the effects of iPRF on macrophage polarization (Figure 4), despite the presence of sGARP being confirmed in iPRF (Figure 1D). Only expression of the M1 marker, MHCII, was found to be significantly elevated (*p* < 0.05) on M0 macrophages treated with iPRF and a blocking anti-GARP antibody, supporting that iPRF polarizes macrophages to a “M0/M2-like” phenotype in a seemingly GARP independent manner. However, it is important to note that Hahn et al., 2016 treated macrophages with a higher concentration of sGARP (10 µg/mL) and used a different experimental set-up than the current study (macrophages treated with sGARP vs. macrophages cocultured with iPRF +/− anti-GARP antibody) [34].

A possible explanation behind the observed effects of iPRF on macrophage polarization seen in this study may lie within the humoral composition of iPRF. Uchiyama et al., 2021 provided helpful insights into the humoral components of PRP where they determined that leukocyte rich PRP contains increased levels of the M2 macrophage polarizing agents, IL-10, TGF-β, M-CSF, and IL-1RA [18]. Varela et al., 2019 could also show an upregulation of TGF-β and IL-10 in iPRF in comparison to peripheral blood [38]. As TGF-β and IL-10 are well known to induce M2 macrophage polarization and they are confirmed to be present in iPRF, it is reasonable to speculate that they could be responsible for the observed effects of iPRF on macrophage polarization seen in this study. A limitation of the present work is that it did not analyze the humoral composition of the iPRF applied to both CD4^+^ T cells and macrophages in detail—rather it focused instead on the phenotypic and functional alterations of immune cells following iPRF application. Future studies are needed to gain further insights into the humoral composition of iPRF and how it may alter macrophage polarization and function.

A study by Zhang et al., 2020 reported that GARP is selectively expressed on the surface of “M1-like” inflammatory macrophages, where it is thought to regulate the release of active TGF-β1 and plays an important role in macrophage shift [39]. They found that pancreatic ductal adenocarcinoma cells directly induce DNA methylation of different genes in “M1-like” macrophages via GARP and integrin αV/β8 binding, which led to a functional reprogramming of macrophages from “M1-like” to “M2-like” macrophages. Blocking of either GARP or integrin αV/β8 could suppress this reprogramming [39]. It remains unclear if iPRF, or more specifically the platelets in iPRF, act upon macrophages in a similar manner, and this should be a topic of future studies.

As described earlier, GARP was found to exist both as a membrane bound protein enriched on the surface of platelets and as a soluble factor found in iPRF (Figure 1). However, surface GARP and sGARP levels were observed to not parallel each other. It is not well understood if sGARP levels mirror surface GARP levels on platelets, and the time it takes to accumulate significant levels of sGARP in the extracellular space. Recent work by Zimmer et al., 2020 has shown that pre-activated platelets exhibit an upregulation of surface GARP expression and an increase of sGARP in their supernatants following 16 h of incubation [24]. A limitation of this work is that sGARP levels in iPRF were only measured directly after its preparation. It could be that platelets only had enough time to upregulate GARP on their surfaces (Figure 1A,B) but not enough time to secrete sGARP in significant quantities. Others have shown that iPRF induced fibrin clots slowly reduce growth factors over time, which could also be true for sGARP, but this must be evaluated in future studies [13].

Metelli et al., 2020 have recently found that thrombin cleaves GARP from the surface of platelets, generating its soluble form (sGARP), and in turn aids in the activation of TGF-ß1 [25]. As thrombin is present in iPRF, it is reasonable to speculate that both sGARP and TGF-ß1 accumulate in iPRF through thrombin mediated proteolytic cleavage, but this should be confirmed in future studies. Of note, Treg are known to also release sGARP from their membranes into the extracellular space, which can induce Treg [34]. As iPRF derived platelets/iPRF were shown to induce a regulatory phenotype in CD4^+^ cells (Figure 2 and Figure 3) in a GARP dependent fashion, it is reasonable to speculate that the resulting Treg are capable of inducing further Treg themselves in a paracrine sGARP dependent manner—thereby amplifying the effects of iPRF derived platelets/iPRF on CD4^+^ cells. However, a more detailed examination into the paracrine effects of GARP from Treg is needed.

The findings of this work also contribute to the understanding of the complicated and often contested interplay between GARP and Foxp3 [20,40,41,42,43]. It was observed that expression of Foxp3 appears to be dependent on GARP. Hereby, GARP, expressed by platelets in iPRF, was able to strongly induce expression of Foxp3 in CD4^+^ T cells, whereas application of anti-GARP antibodies could reverse these effects back to untreated CD4^+^ T cell levels (Figure 2A and Figure 3A). Despite reduced expression of Foxp3 upon anti-GARP antibody treatment, GARP levels remain unaffected further supporting that the expression of GARP is not dependent on Foxp3 (Figure 2B). Notably, blocking with an anti-TGF-β-1-III antibody or an anti-TGF-β receptor II antibody were unable to significantly reduce Foxp3 expression (Figure 3A). These findings are in accordance with the literature where the role of platelets in the induction of Treg via a GARP-dependent mechanism have been already described [24]. Furthermore, it has been found that silencing of GARP partially impairs the suppressive capacity of Treg. In a murine model, lacking surface GARP on Tregs, due to a Treg-specific deletion of the chaperone molecule GP96, mice developed fatal multi-organ inflammatory disease [44]. This is in agreement with the present findings as the expression of the Treg master regulator, Foxp3, and the associated suppressive properties of Treg, including inhibition of effector cytokine production (IL-2 and IFN-γ), were observed to be highly dependent on surface GARP expression (Figure 2 and Figure 3). Collectively, this indicates that the expression of Foxp3 and associated suppressive properties of Treg are highly dependent on GARP.

A possible application of this work could be the addition of sGARP to APC to further enhance their wound healing properties, namely through the induction of Treg, as observed in this study. sGARP has already been shown to exhibit potent anti-inflammatory properties on human CD4^+^ T cells, including the induction of suppressive Treg and an inhibition of T effector cells [34,45]. These sGARP mediated effects are in accordance with the observed GARP dependent regulatory effects on CD4^+^ T cells in the present work. Whether the presence of sGARP, GARP expressed on the surface of platelets, or a mixture of both, are responsible for the observed regulatory effects of iPRF on CD4^+^ T cells remains unclear. Of note, Hahn et al., 2013 showed that sGARP treatment could prevent the induction of graft-versus-host disease (GVHD) in a humanized mouse model of xenogeneic GVHD via the simultaneous accumulation of Treg and the inhibition of destructive T effector cells [45]. Altogether, these potent anti-inflammatory properties of sGARP indicate its strong potential to boost the wound healing properties of APC. However, future studies are greatly needed to thoroughly evaluate the efficacy and safety of the addition of sGARP to platelet concentrates. Additionally, in the future, great care should be exercised when selecting patients that could benefit from this treatment as the observed immunological effects of iPRF/sGARP could be problematic for cancer patients.

## 5. Conclusions

The distinctive interplay of autologous platelet concentrates with CD4^+^ T cells and macrophages were analyzed in this study. A GARP dependent regulatory influence of platelets derived from iPRF on CD4^+^ T cells was shown for the first time. Furthermore, application of iPRF could polarize human monocyte derived macrophages to a “M0/M2-like” wound healing phenotype in a GARP independent manner. Collectively, this indicates that the immunological capacity of iPRF is partially mediated by GARP, especially in terms of its effects on CD4^+^ T cells. The findings from this work may help to better understand the immunological reactions after iPRF application during the wound healing process. Furthermore, it may be possible to therapeutically enhance the wound healing properties of APC further via the addition of a sGARP to new concentrates.

## Figures and Tables

**Figure 1 biomedicines-10-03136-f001:**
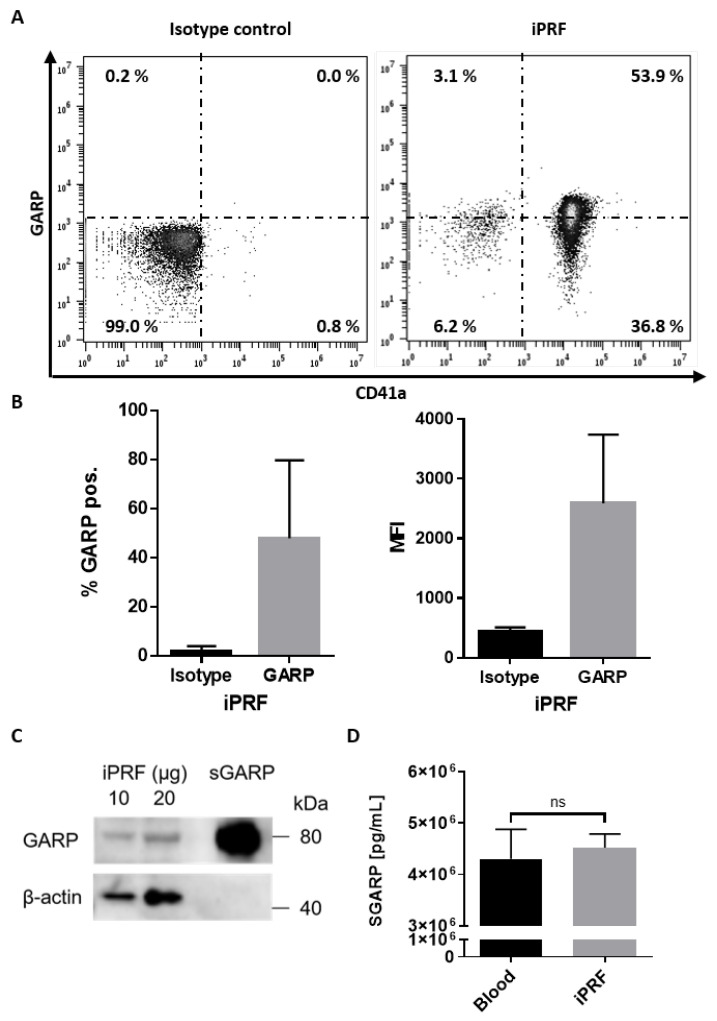
Glycoprotein A repetitions predominant (GARP) is expressed on the surface of platelets and exists as a soluble factor found in liquid platelet rich fibrin (iPRF). (**A**,**B**) iPRF was stained directly for the platelet marker, CD41a, and GARP and underwent flow cytometric analysis. Isotype antibodies were used as a control. (**A**) Dot blots show one representative result. (**B**) Bar diagrams of GARP expression show the pooled data of percentages (%) of GARP positive platelets in iPRF and the mean fluorescence intensity (MFI) of GARP (n = 3 ± SD, analyzed by paired Students *T*-tests). (**C**) Expression of GARP in iPRF was analyzed via Western blot. iPRF lysate was loaded in increasing amounts. Recombinant soluble GARP (sGARP) was loaded at 1 ng/lane and served as a positive control, whereas β-actin acted as the loading control. (**D**) Bar graph shows the soluble GARP (sGARP) levels in peripheral blood and iPRF analyzed via ELISA (n = 3 ± SD, ns determined by paired Students *T*-tests).

**Figure 2 biomedicines-10-03136-f002:**
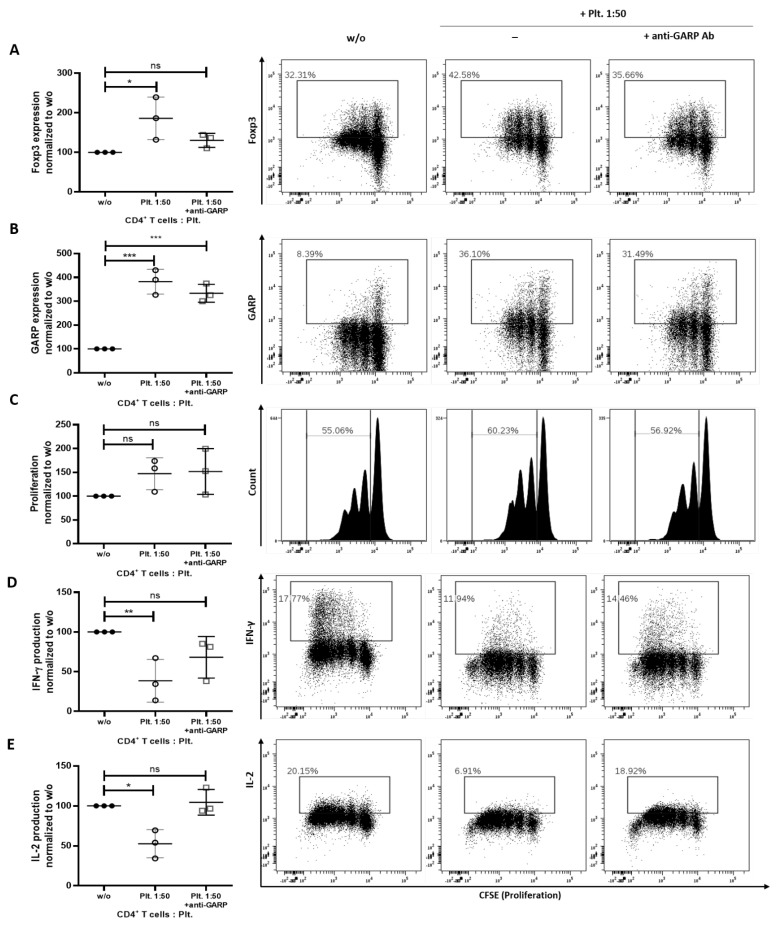
Isolated platelets from liquid platelet rich fibrin (iPRF) induce a regulatory phenotype in CD4^+^ T cells. CD4^+^ T cells and platelets (Plt.) were cocultured at a ratio of 1 to 50 and were treated with 10 µg/mL anti-GARP antibody (Ab). CD4^+^ T cells were prelabled with carboxyfluorescein succinimidyl ester (CFSE) to track their proliferation and stimulated with 0.5 µg/mL anti-CD3 and 1.0 µg/mL anti-CD28 Abs in the presence or absence of platelets. (**A**–**C**) The expression of Foxp3 (**A**), GARP (**B**), and proliferation (**C**) of CD4^+^ T cells was measured on day 3 of coincubation via flow cytometry. (**D**,**E**) After 6 days of coincubation, cells were stimulated with phorbol 12-myristate 13-acetate (PMA), ionomycin, and monesin for 5 h. IL-2 and IFN-γ production was quantified via intracellular flow cytometry. Representative dot plots of three independent experiments are shown. Graphs show CD4^+^ T cells cocultured with platelets normalized to untreated CD4^+^ T cells alone control (n = 3, means ± SD, * *p* < 0.05, ** *p* < 0.01, *** *p* < 0.001, and ns determined by one-way ANOVA).

**Figure 3 biomedicines-10-03136-f003:**
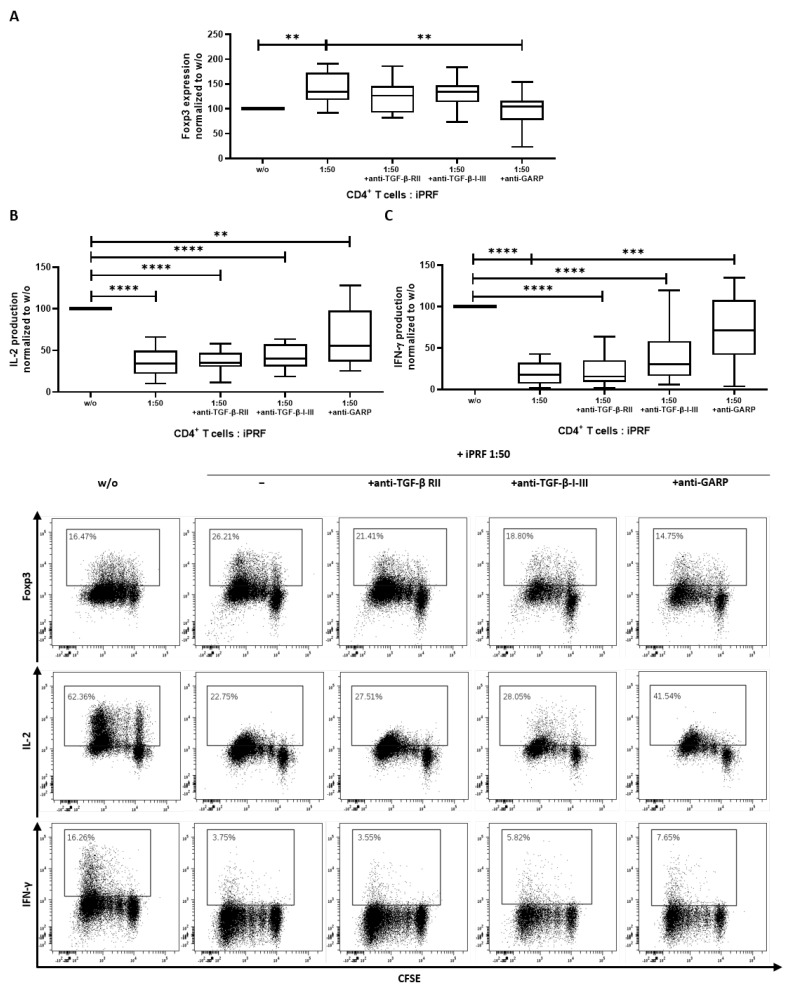
Liquid platelet rich fibrin (iPRF) induces a regulatory phenotype in CD4^+^ T cells. CD4^+^ T cells and iPRF were cocultured at a ratio of 1 to 50 and treated with either 10 µg/mL anti-TGF-ß receptor II (anti-TGF-β-RII) antibody (Ab), 10 µg/mL anti-TGF-β-1-III Ab, or 10 µg/mL anti-GARP Ab. CD4^+^ T cells were prelabled with carboxyfluorescein succinimidyl ester (CFSE) to track their proliferation and stimulated with 0.5 µg/mL anti-CD3 and 1.0 µg/mL anti-CD28 Abs in the presence or absence of iPRF. (**A**) The expression of Foxp3 was measured on day 3 of coincubation using intranuclear flow cytometry. (**B**,**C**) After 6 days of coincubation, cells were stimulated with phorbol 12-myristate 13-acetate (PMA), ionomycin, and monesin for 5 h. IL-2 and IFN-γ production was quantified via intracellular flow cytometry. Representative dot plots of 10 independent experiments are shown. Box plots show data which was normalized to the untreated CD4^+^ T cell control (n = 10, box and whiskers, medians ± min/max, ** *p* < 0.01, *** *p* < 0.001, and **** *p* ≤ 0.0001 determined by two-way ANOVA).

**Figure 4 biomedicines-10-03136-f004:**
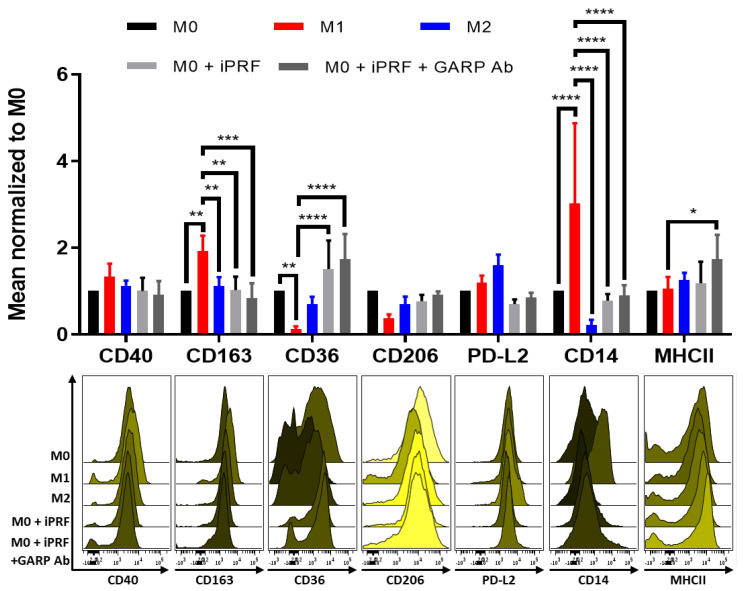
Liquid platelet rich fibrin (iPRF) polarizes M0-macrophages in a more “M0/M2-like” phenotype through a GARP independent mechanism. Human monocyte derived macrophages were polarized, treated with 400 µL iPRF/well, 10 μg/mL anti-GARP antibody (Ab) or were left untreated for 2 days as indicated. Surface marker expression was analyzed via flow cytometry. Bar diagram shows the mean fluorescence intensity (MFI) of each surface marker normalized to the respective M0 control. Representative histograms of 3 independent experiments are shown (n = 5 donors, bar means ± SD, * *p* < 0.05, ** *p* < 0.01, *** *p* < 0.001, and **** *p* < 0.0001 determined by two-way ANOVA).

## Data Availability

The data presented in this study are available in this article and in the Appendix A.

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
