# Peer review of "GARP Regulates the Immune Capacity of a Human Autologous Platelet Concentrate"

_biomedicines, 2022, doi:10.3390/biomedicines10123136_

Round 1

Reviewer 1 Report

In this manuscript, the Authors deal with an autologous platelet concentrate - liquid platelet rich fibrin (iPRF), that is known for its ability to ameliorate and fasten wound healing. In detail, Authors analyze the interaction of iPRF with immune cells, like monocyte derived macrophages and CD4+ T cells, implicated in the wound healing process with particular attention on the influence of GARP. The results suggest that the immune capacity of iPRF is mediated in part by GARP and by its ability to induce regulatory CD4+ T cells.

Even if a lot of papers are published regarding the abilities of platelet concentrates to ameliorate wound closure, the real mechanisms understanding the wound healing properties of platelets are not completely deciphered. The role of GARP in these processes has never been described.

My comments about the manuscript are positive, I have only some minor concerns:

-        Methods lines 143-144. It is not clear what Authors mean when they state “CD4+ T cells and macrophages in coculture with iPRF”, please clarify the term “coculture” (also line 201)

-        Results line 250: why “approximately half the platelets expressed GARP”? Are there different type of platelets, or are platelets functionally different? Or the number of positive platelets is somewhat related to GARP cleavage from platelet membrane? Please clarify

-        As to the ability of iPRF to polarize macrophages to an “M2 like” phenotype, and to the participation of GARP to this effect, I think that this is a crucial point because of the pro-carcinogenic effect of M2 macrophages: if sGARP might be added to platelet concentrates to therapeutically enhance the wound healing properties of APC, its safety need to be completely determined.

Reviewer 2 Report

This study focused on the possible immune capacity of platelet concentrates and investigated GARP’s role in the conversion of macrophages and T cells. The role of GARP in PRP used for regenerative therapy has not been well studied and is poorly understood. Thus, this is an interesting challenge. This reviewer appreciates what was done in this study. However, it is difficult to understand why the authors designed these experiments.

1)        Based on many basic and clinical reports, it has generally been thought that leukocyte-rich PRP (L-PRP) exacerbates inflammation, but does not facilitate tissue regeneration by accelerating the exclusion of dead cells and that debris. To minimize such inflammation and facilitate tissue regeneration, as Anitua has repeatedly claimed, it is necessary to exclude white blood cells from PRP preparations. Because of his clinical experiences, this reviewer agrees with his concept. Thus, if the authors postulated white blood cells included in PRP preparations, this reviewer does not understand what they thought. Alternatively, did the authors assume the added T cells and macrophages as infiltrated ones in vivo? If so, the authors should first show their clinical questions and clearly mention the study’s rationale and purpose.

2)        This reviewer wonders why the authors chose iPRF instead of P-PRP. Does citrate influence GARP’s action?

3)        Because of the short spin at a relatively low centrifugal force, iPRF may not include a significant number of WBC (Please disclose the CBC data). Instead, because of no added anticoagulants, recognizable amounts of fibrin fibers appear within several 10 minutes even in plastic tubes, and platelets aggregate with one another, even though the sample is placed without mechanical stimuli, like centrifugation. In Line 140, the authors described that platelets in the pellets were counted. However, it seems practically difficult (or almost impossible) to resuspend platelets if the donors were healthy enough.

4)        The data of the donors (number, age, gender, medication, etc.) should be disclosed around Line 135. In addition, prior to the subsection of Flow cytometry, isolation of macrophages and T cells should be placed. Otherwise, in Line 143, readers cannot smoothly follow the sudden appearance of T cells and macrophages.

5)        In the subsection of Flow cytometry, this reviewer wonders if their manual disruption of the fibrin clot may injure blood cells included there (Line 145). Please describe the procedures for efficient retention of the cytokines and the transcription factor inside the cells in more detail.

6)        In the abstract, the authors mentioned that platelets highly express GARP on their surface. However, in Figure 1, only less than 50% of platelets on average expressed GARP. What does it imply? In addition, Panel D shows that although platelets are the main source of GARP, GARP is not concentrated in iPRF. Why isn’t platelet-derived GARP concentrated in platelet concentrates?

7)        When the scatter pots shown in Figure 1 are compared with those in Figure 2, the major cell populations seem different: In Figure 1, especially panels shown in the upper two rows, the majority seems D0, while in Figure 2, the majority generally seems D4 (and maybe D5), which were composed of highly proliferating cells. Please explain this difference.

8)        As described somewhere, GARP is expressed also in macrophages. If so, the neutralizing antibody of GARP may block not only platelet-derived GARP but also macrophage-derived GARP. How about T cells? How about the involvement of GARP derived from these cells in their induction in an autocrine manner?

9)        In Line 48, the authors described that thrombin and calcium chloride are anticoagulants. It is correct?

10)    The introduction section is too long and not well-focused.

Round 2

Reviewer 2 Report

Well done.